# Altered thymic differentiation and modulation of arthritis by invariant NKT cells expressing mutant ZAP70

Meng Zhao[1], Mattias N.D. Svensson[2,3], Koen Venken[4,5], Ashu Chawla[6], Shu Liang[7], Isaac Engel[1], Piotr Mydel[8,9], Jeremy Day[10], Dirk Elewaut[4,5], Nunzio Bottini[2,3] & Mitchell Kronenberg [1,11]

Various subsets of invariant natural killer T (iNKT) cells with different cytokine productions develop in the mouse thymus, but the factors driving their differentiation remain unclear. Here we show that hypomorphic alleles of Zap70 or chemical inhibition of Zap70 catalysis leads to an increase of IFN-γ-producing iNKT cells (NKT1 cells), suggesting that NKT1 cells may require a lower TCR signal threshold. Zap70 mutant mice develop IL-17-dependent arthritis. In a mouse experimental arthritis model, NKT17 cells are increased as the disease progresses, while NKT1 numbers negatively correlates with disease severity, with this protective effect of NKT1 linked to their IFN-γ expression. NKT1 cells are also present in the synovial fluid of arthritis patients. Our data therefore suggest that TCR signal strength during thymic differentiation may influence not only IFN-γ production, but also the protective function of iNKT cells in arthritis.

[1] Division of Developmental Immunology, La Jolla Institute for Allergy and Immunology, La Jolla, CA 92037, USA. [2] Division of Cellular Biology, La Jolla Institute for Allergy and Immunology, La Jolla, CA 92037, USA. [3] Department of Medicine, University of California, San Diego, La Jolla, CA 92037, USA. [4] Department of Internal Medicine, Faculty of Medicine and Health Sciences, Ghent University, De Pintelaan 185, B-9000 Ghent, Belgium. [5] VIB Inflammation Research Center, Ghent University, B-9000 Ghent, Belgium. [6] Bioinformatics Core Facility, La Jolla Institute for Allergy and Immunology, La Jolla, CA 92037, USA. [7] Division of Vaccine Discovery, La Jolla Institute for Allergy and Immunology, La Jolla, CA 92037, USA. [8] Department of Clinical Science, University of Bergen, Bergen N-5021, Norway. [9] Department of Microbiology, Jagiellonian University, Krakow 30-387, Poland. [10] NGS facility, La Jolla Institute for Allergy & Immunology, La Jolla, CA 92037, USA. [11] Division of Biological Sciences, University of California San Diego, La Jolla, CA 92037, USA. These authors contributed equally: Meng Zhao, Mattias N. D. Svensson. These authors Jointly supervised: Nunzio Bottini, Mitchell Kronenberg. Correspondence and requests for materials should be addressed to N.B. (email: nbottini@ucsd.edu) or to M.K. (email: mitch@lji.org)

nvariant natural killer T cells (iNKT cells) are a population that expresses an invariant TCR α chain, encoded by a *Trav11-Traj18* rearrangement in mice. They also express a limited set of *Trbv* genes, although unlike for the α chain, there is substantial CDR3β diversity as a result of gene rearrangement. iNKT cells recognize glycolipid antigens from microbial, environmental and even autologous sources when presented by CD1d, a non-polymorphic class I-like antigen presenting molecule[1]. Their rearranged α chain is conserved in a number of mammalian species, including humans, as is the specificity of iNKT cells[2], suggesting they have an important function. An unusual property of iNKT cells is their innate-like responses, typified by the rapid secretion of large amounts of cytokines after TCR stimulation[3]. They also can respond rapidly when exposed to certain cytokines, similar to NK cells and other ILC populations. For example, the combination of IL-12 and IL-18 will stimulate IFN-γ synthesis by iNKT cells[4,5].

iNKT cells have been reported to influence many types of immune responses, including chronic inflammatory conditions and autoimmunity[6]. A puzzling feature, however, is that in some cases iNKT cells stimulate immunity and inflammation and have a detrimental influence, while in other situations they can be anti-inflammatory and beneficial. Considering rheumatoid arthritis (RA) models induced by immunization, for example collagen-induced arthritis (CIA), or induction by serum transfer, in most studies iNKT cells had a disease promoting effect. Therefore, mice without iNKT cells were better off in several studies, although this was not true in some other reports[7]. Their role in patients is unproven, but iNKT cells consistently are decreased in the peripheral blood of those with RA[6].

One possible contributor to the divergent effects of iNKT cells is that there could be a selective differentiation in the thymus or selective expansion or peripheral activation of functional subsets of these cells. Subsets of iNKT cells include NKT1, NKT2, and NKT17 cells, analogous to CD4$^+$ $T_H1$, $T_H2$, and $T_H17$ cells, respectively[8]. An important difference with CD4$^+$ T lymphocytes, however, is that these functional subsets differentiate in the thymus. The mechanism is unknown that causes iNKT cells, with their limited TCR diversity and highly similar specificities, to differentiate in the thymus to specialized effector populations with highly different transcriptomes.

To investigate the possible influence of TCR strength in the differentiation of a polyclonal iNKT cell population, we analyze two strains of mice in which the function of the ζ-chain-associated protein kinase 70 (ZAP70) is reduced. ZAP70 is a tyrosine kinase that phosphorylates the linker of activated T cells (LAT) and the SH2 domain-containing leukocyte protein of 76 kDa (SLP-76), and therefore has an important role in early TCR signaling events. One of the strains we analyze is the SKG mouse, which has a spontaneous mutation in an SH2 domain of ZAP70 that generates a hypomorphic allele. SKG mice have altered thymic selection leading to the generation of arthritogenic Th17 cells[9,10]. The other strain (ZAP70AS) has an analog-sensitive(AS) allele of ZAP70[11].

In analyzing these mice, we demonstrate that ZAP70 influences not only the number of iNKT cells, but also affects the proportions of iNKT cell subsets and the way in which they function. Furthermore, using the SKG mice, we demonstrate that iNKT cells are capable of localizing to synovial tissue and preventing severe arthritis, in part through production of IFN-γ. Therefore, understanding the role of different iNKT cell subsets could help design new therapeutic interventions for rheumatoid arthritis patients.

## Results

**TCR signaling genes differ in iNKT cell subsets**. The different patterns of iNKT cell cytokine secretion could be critical for

understanding the diverse functions that have been attributed to these cells. In order to understand how functional subsets of iNKT cells arise, we evaluated the transcription programs in subsets of thymus iNKT cells[12]. Because TCR avidity plays an important role in the development of CD4 and CD8 cell populations, as well as agonist selected iNKT cells, we analyzed if the expression of genes related to the TCR signaling pathway differed among iNKT cell subsets. A Gene Set Enrichment Analysis (GSEA) of the RNA-seq results revealed that TCR signaling genes were significantly enriched in NKT1 thymocytes compared to NKT2 cells or NKT17 (Fig. 1a; Supplementary Fig. 1a), and were significantly enriched in NKT17 compared to NKT2 cells (Supplementary Fig. 1a). These data indicate that iNKT cell subsets activated different transcriptional programs for genes in the TCR signaling pathway after their generation in the thymus. We also observed the highest TCRβ staining as well as antigen-loaded CD1d tetramer staining in NKT2 cells, intermediate in NKT17 and lowest in NKT1 (Supplementary Fig. 1b), consistent with a previous publication[13]. We grouped the differentially expressed (DE) genes (adjusted $p <= 0.1$, fold change $>= 1.5$) according to relative expression levels in the subsets (Fig. 1b; Supplementary Fig. 1c). Several genes in the NF-κB pathway, including *Rela*, *Bcl10*, *Malt1* and inhibitors *Nfkbib* and *Cblb* were among the NKT1 enriched genes (Fig. 1b). This was consistent with the selective defect in CD44$^{hi}$NK1.1$^+$ (stage 3) iNKT cells, which are mostly NKT1 cells, in IκBα$^{tg}$[14] and PKCθKO[15] mouse strains that both have impaired NF-κB signaling. It has been shown that PLZF was induced during thymic development by Egr2 downstream of NFAT signaling[16]. Expression of *Egr2* and *Nfatc1* was highest in NKT2 cells. *Egr1* and *Nfatc3* were preferentially expressed in NKT17 cells, but also higher in NKT2 compared to NKT1 cells. Several activation markers typically associated with a strong interaction of TCR and cognate antigens, such as *Icos*[17] and *Pdcd1*[18] were also more highly expressed in NKT2 and NKT17 cells, or selectively in NKT2, such as *Cd5*[19]. Conversely, negative regulators for TCR signaling, including, *Ptpn22*[20], *Pag1*[21], *Ptpn6*[22], *Ptprc*[23], *Lat2*[24], and *Ubash3a*[25,26] were more highly expressed in NKT1 cells (Fig. 1b). These results suggest that unique gene signatures in iNKT cell subsets were induced during development. Based on this, we hypothesized that the development or maintenance of NKT1 cells required a lower strength of TCR signals.

**Reduced iNKT cells in ZAP70 mutant mice**. To test the signal strength hypothesis further, we analyzed SKG mice, which harbor a spontaneous point mutation in the *Zap70* gene that results in reduced TCR signals, defective thymic selection of autoimmune T cells and eventual development of inflammatory arthritis dependent on IL-17 secretion[9]. The percentages of iNKT cells in CD8$^-$ thymocytes in SKG and WT BALB/c control mice were comparable (Fig. 1c), but in agreement with a previous report[9], there was decreased positive selection of CD4 and CD8 T cells in these mice. Therefore, the percentage of iNKT cells in total live thymocytes and the number of iNKT cells in SKG thymus were significantly lower (Fig. 1d, e).

We carried out several experiments to determine the cause of the reduced iNKT cell frequency. The expression of CD1d by SKG DP thymocytes, the cell type that positively selects iNKT cells, was not reduced (Supplementary Fig. 1d), suggesting the decrease could be cell intrinsic. Mutations that affect the survival time of DP thymocytes can cause a cell-intrinsic decrease in iNKT cells, due to the decreased formation of the iNKT cell TCR α chain[27,28]. This is because the *Trav11* and *Traj18* gene segments in the α locus are in distal locations and their joining

in DP cells usually requires the secondary TCR α locus rearrangements dependent on normal DP survival. The average number of DP cells in 8-week-old mice, however, was comparable in WT and SKG strains (Supplementary Fig. 1e). Furthermore, the rate of incorporation of BrdU into the newly synthesized

DNA of DP thymocytes was similar comparing SKG and WT mice (Supplementary Fig. 1f). Using AnnexinV (AnV) staining to measure apoptotic cells, there were decreased AnV$^+$ cells in SKG mice (Supplementary Fig. 1g), implying that DP cells may survive even longer than in WT mice.

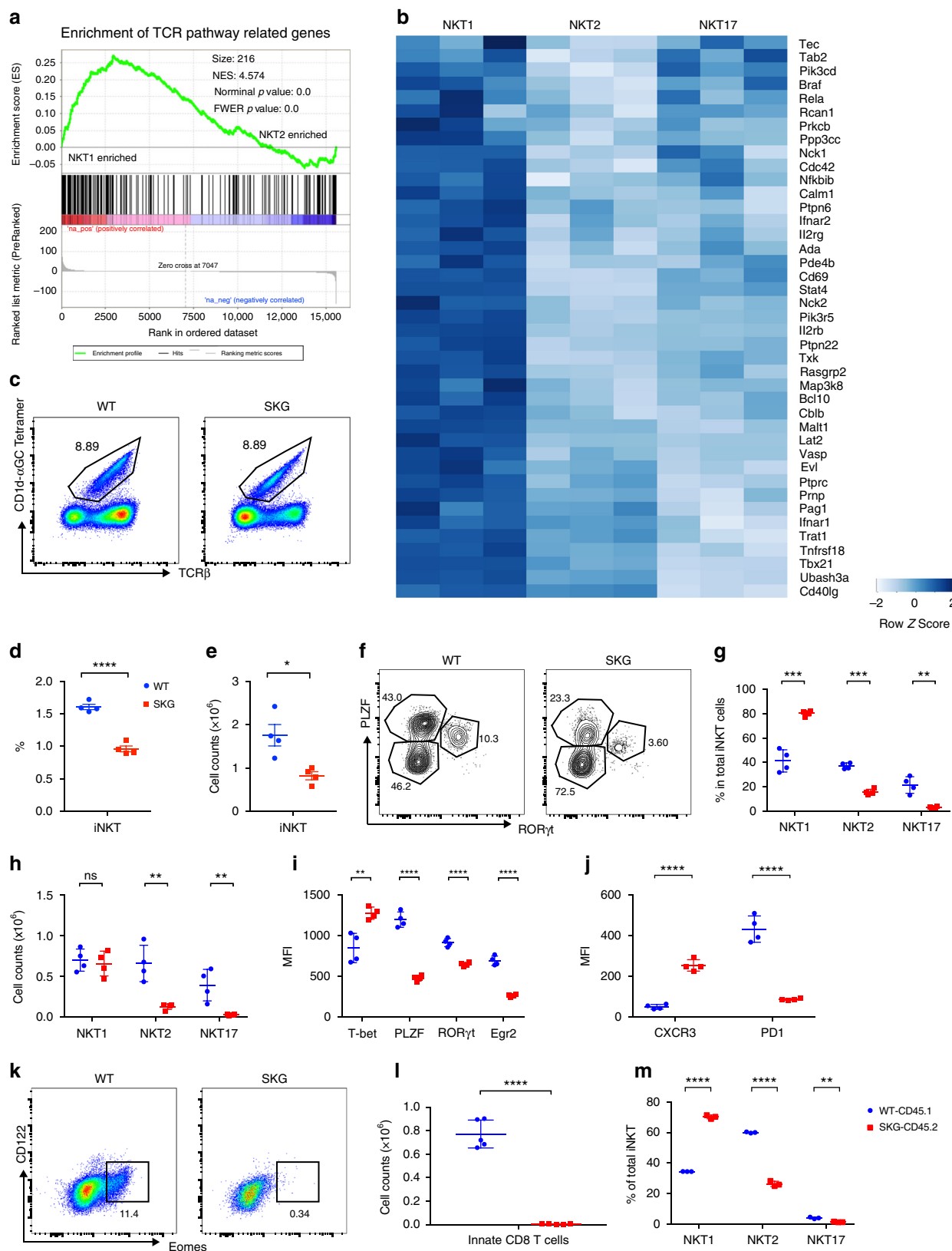

To determine if TCR α locus rearrangements were skewed away from distal gene segments, we examined the *Traj* usage of *Trav11* rearrangements in CD69−, CD5− pre-selection DP thymocytes by sequence analysis. As controls, we also analyzed rearrangement of *Trav19*, which is one of the most *Traj* proximal Vα segments (Supplementary Fig. 2a), and *Trav9d-3* located towards the distal region. *Trav9* has been shown to rearrange much less frequently to distal *Traj* genes in HEB knockout mice, which have impaired DP survival[27]. Considering productive reads, *Trav19* preferentially used the proximal *Traj* genes in BALB/c and SKG mice (Supplementary Fig. 2b), consistent with results from pre-immune CD8 T cells[29], while *Trav9d-3* used distal ones (Supplementary Fig. 2c). The *Trav11-Traj18* rearrangement accounted for ~10% of total reads from *Trav11* amplicons in SKG mice, the same frequency as in WT mice (Supplementary Fig. 2d), showing that the decrease of iNKT cells (Fig. 1d, e) was not due to reduced generation of the invariant TCRα chain rearrangement. Comparing *Traj* rearrangements in WT and SKG mice for all three *Trav* genes, a significant bias for distal *Traj* rearrangements in SKG DP thymocytes was observed, (Supplementary Fig. 2e–g). The hypomorphic *Zap70* mutation in SKG mice dampened TCR signaling in DP cells, as reflected by reduced free calcium flux into the cytoplasm upon TCR stimulation (Supplementary Fig. 2h). Therefore, it is possible that generation of a TCR that passes the threshold of positive selection may require more rounds of TCRα rearrangement in the DP cells from SKG mice, leading to a higher frequency of rearrangements to the distal *Traj* segments. This may compensate for the reduced TCR signal strength, limiting the decrease in the iNKT cell population in these mice.

**Skewed iNKT cell subsets in *Zap70* mutant mice.** The reduction in the MFI of TCRβ and CD1d-αGC tetramer staining (Fig. 1c), suggested that NKT1 cells could be more abundant in thymic SKG iNKT cells. Staining for transcription factor expression in thymic iNKT cells confirmed the enrichment of NKT1 cells (PLZF$^{lo}$RORγt$^-$) in SKG mice (Fig. 1f), whereas NKT2 and NKT17 cells were much reduced (Fig. 1f–h). Consistent with the change in the subsets, T-bet expression in total iNKT cells was much higher, and PLZF and RORγt were diminished (Fig. 1i). The chemokine receptor CXCR3 has been shown to be most highly expressed by NKT1 cells[12], and its expression was much higher in total SKG iNKT cells. PD-1, a marker for the NKT2/17 subsets (Supplementary Fig. 1c) had reduced expression in SKG iNKT cells (Fig. 1j). NKT2 cells capable of producing IL-4 under homeostatic conditions have been shown to be critical for the differentiation of innate, memory-like CD8 T cells that express Eomes instead of T-bet, and a high level of CD122[30]. Consistent with the relatively few NKT2 cells in SKG thymus, innate memory CD8 T cells were barely detectable in these mice (Fig. 1i, k).

When stimulated directly ex vivo on microwells coated with mouse CD1d and loaded with αGC, SKG iNKT cells were defective at producing different cytokines, including IFN-γ as well as IL-4 and IL-17 (Supplementary Fig. 3a), likely reflecting the ZAP70-hypomorphic nature of SKG mutation. When using Phorbol 12-myristate 13-acetate (PMA) and ionomycin stimulation to bypass proximal TCR signaling events, however, we observed higher IFN-γ and lower IL-4 and IL-17 production (Supplementary Fig. 3b), likely a result of the increased NKT1 cells in SKG mice. The analysis of mixed bone marrow chimeric mice (Fig. 1m; Supplementary Fig. 3c, d), generated using equal numbers of SKG(CD45.2) and WT(CD45.1) bone marrow donor cells, demonstrated that the defect in iNKT cell subset development was predominantly cell instrinsic, as the SKG iNKT cells of the chimeric mice showed similar defects as in the intact mice.

**Skewed SKG iNKT cell differentiation in organ cultures.** Fetal thymic organ culture (FTOC) is a powerful tool to study T cell development in vitro[31–33]. We tested if iNKT cell subsets would be observed in this system. Analyzing intact mice we confirmed that iNKT cells did not appear in the thymus until several days postnatal age[34]. Therefore, we used E18.5—day 0 animals from BALB/c mice cultured for 1 week in vitro. iNKT cells could be detected from day 4 in culture and their percentage in total thymocytes gradually increased in the following days (Fig. 2a, b). On day 4, iNKT cells were mostly PLZF$^{hi}$ T-bet$^{lo}$ and PD-1$^{hi}$, consistent with the phenotype of iNKT cells from 3-day-old mice in vivo (Supplementary Fig. 4a, b). T-bet expression rapidly increased in the next 24 h. By day 6, the T-bet$^{hi}$ population increased, including cells that retained high expression of the NKT2 marker PD-1, other cells with decreased PD-1 and even some with expression of the NKT1 cell marker CXCR3. Ontogeny studies of iNKT cell phenotype from the thymus of very young animals showed a similar phenotypic progression (Supplementary Fig. 4a, b). We did not detect RORγt expression in the organ cultures and NKT17 cells were a very minor population in 3-day-old or 4-day-old mice. We also cultured thymus tissue from *Nur77$^{gfp}$* mice, which express a reporter for TCR signal strength[35]. Total iNKT cells, as reported, showed very low GFP signal compared to CD4 T cells, but stage 0 iNKT cells (CD24$^{hi}$ CD44$^-$ NK1.1$^-$) expressed high level of GFP (Supplementary Fig. 4c, d). In FTOC, we observed a higher GFP signal from the PD1$^{hi}$NK1.1$^-$ population that resembles NKT2 cells (Fig. 2c, d). These *Nur77$^{gfp}$* high expressing NKT2 cells have been reported to produce IL-4 in vivo at steady state[36]. This is also consistent with the higher transcription of *Nfatc1*, *Egr2*, and *Pdcd1* in NKT2 cells, as shown in Supplementary Fig. 1c. We compared cultures of thymus tissue from day 0 SKG and WT animals. The SKG iNKT cells in the culture expressed a higher amount of CXCR3 and lower PD-1 (Fig. 2e), in line with the NKT1 skewed phenotype of SKG mice. Therefore, overall FTOC cultures recapitulated some important features of iNKT cell development and the subset alterations in SKG mutant mice, indicating that

**Fig. 1** iNKT cells are skewed to NKT1 phenotype in SKG ZAP70 mutant mice. **a** Gene set enrichment analysis of 216 TCR signaling pathway-related genes comparing the transcriptomes of thymic NKT1 and NKT2 cells from BALB/c mice. **b** Differentially expressed TCR pathway genes enriched in NKT1 cells from triplicate RNA-seq runs are shown in a heatmap. **c–e** Decreased thymic iNKT cells in SKG mice. **c** Representative staining of iNKT cells gated as TCRβ$^{int}$, CD1d-αGC tetramer$^+$ cells in live, CD8-negative thymocytes from WT and SKG mice. **d** Percentages and **e** absolute numbers of thymic iNKT cells in the WT and SKG mice. **f** iNKT cell subsets were gated as PLZF$^{lo}$ RORγt$^-$ (NKT1), PLZF$^{hi}$ RORγt$^-$ (NKT2), and PLZF$^{int}$RORγt$^+$ (NKT17). **g** Percentages in total iNKT cells and **h** absolute numbers of the iNKT cell subsets in WT and SKG mice. **i, j** Expression of the indicated transcription factors and surface markers CXCR3 (NKT1 marker) and PD-1 (NKT2) marker in total WT or SKG iNKT cells. **k** Innate-like CD8 T cells, gated as Eomes$^+$ CD122$^+$ in CD8$^+$CD3ε$^{hi}$ thymocytes. **l** Absolute numbers of innate-like CD8 T cells in WT and SKG mice. **m** Thymic iNKT cell subsets from mixed bone marrow chimera mice. Percentages of iNKT subsets from either WT (CD45.1) or SKG (CD45.2) donors were plotted. **c–m** Data are representative from one of at least three independent experiments. Graphs represent mean ± SD with symbols representing individual mice. *$p < 0.05$; **$p < 0.01$; ***$p < 0.001$; ****$p < 0.0001$; n.s. not significant (unpaired two-tailed Student's *t* test)

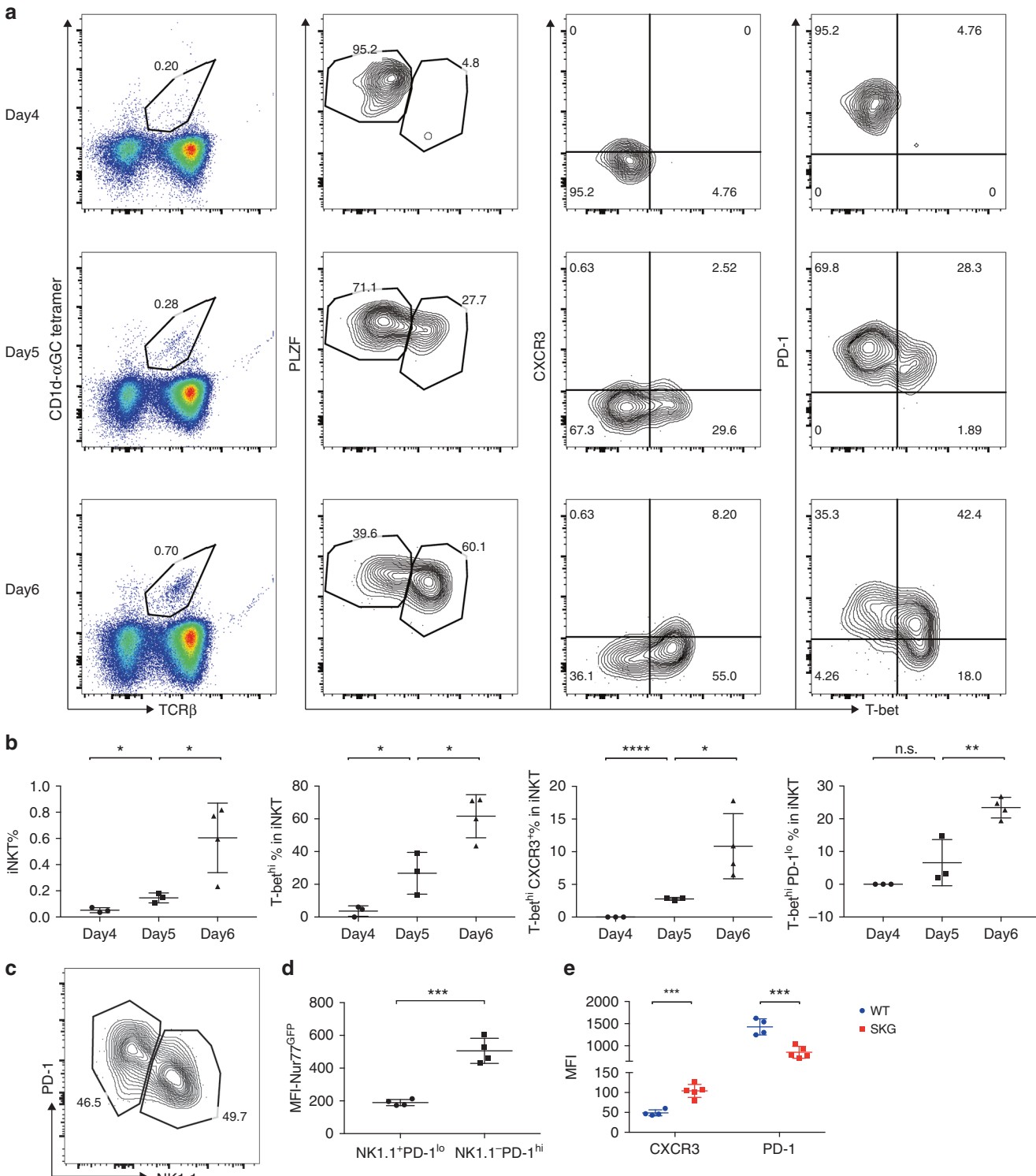

**Fig. 2** The development of iNKT cell subsets in thymic organ culture. **a, b** Thymic lobes from E18.5—day 0 BALB/c embryos were cultured in vitro and examined on days 4, 5, and 6. Percentages of total iNKT cells (TCRβ$^{int}$, αGC tetramer$^+$) in live cells are shown (left column). Expression of transcription factors T-bet and PLZF (second column), surface markers CXCR3 and PD-1 (third and fourth columns) in total iNKT cells are shown. **b** Compiled data of the experiments. **c** Thymic lobes from day 0 *Nur77$^{gfp}$* mice were cultured in vitro as in (**a**). **d** Total iNKT cells were separated into NK1.1$^+$PD-1$^{lo}$ (NKT1-like) and NK1.1$^-$PD-1$^{hi}$ (NKT2-like) populations and MFI of GFP from the two populations were plotted. **e** Thymic lobes from day 0 SKG and WT control mice were cultured in vitro as in (**a**), and expression levels of CXCR3 and PD-1 were compared. Data are representative of three independent experiments. Graphs represent mean ± SD with symbols representing individual mice. *$p < 0.05$; **$p < 0.01$; ***$p < 0.001$; ****$p < 0.0001$; n.s. not significant (unpaired two-tailed Student's *t* test)

migration of mature T cells back to the thymus cannot explain the iNKT cell phenotype in the thymus of SKG mice.

**Skewed iNKT cell development in ZAP70AS mice.** The SKG mutation in the second SH2 domain of ZAP70 affects its binding to phosphorylated ITAMs. To examine more directly the effect of the kinase activity of ZAP70 in lowering the strength of TCR signaling during iNKT cell development, we utilized mice expressing an engineered, analog-sensitive ZAP70 (ZAP70AS), whose catalytic activity can be specifically targeted by small molecule inhibitors[37]. Although ZAP70AS is a hypomorphic mutant, with a catalytic activity approximately one third of the WT protein[37], it has been shown that mice that expressed this allele reconstituted a normal T cell population in ZAP70 null mice[11]. Because AS mice were on the C57BL/6 background, which is NKT1 biased[36], NKT1 cells were the major subset in this strain (60% of total iNKT cells in WT mice, Fig. 3a). Interestingly, similar to the comparison of SKG mice with their parental BALB/c strain, NKT1 cells in AS mice were more prevalent (Fig. 3a, b). In agreement with this, we observed changes in expression of the transcription factors T-bet and PLZF, as well as subset markers CXCR3 and PD-1, that were in an NKT1-biased direction in AS and SKG mice when compared to their WT controls (Figs. 1i, j and 3c, d).

**Inhibition of ZAP70 kinase decreases NKT2 cells.** We tested if small molecule inhibition of ZAP70 catalysis in FTOC would further affect iNKT cells (Fig. 3e–h). Thymus lobes from day 0 animals were cultured in the presence of DMSO or the inhibitor 3-MB-PP1. After 7 days of culture, T lymphocytes including iNKT cells were greatly diminished, even at the lowest concentration tested (Supplementary Fig. 5a–c). This is in line with the requirement for the ZAP70 kinase during positive selection[38]. Addition of the inhibitor 48 h after the initiation of the culture, however, allowed for a similar percentage of iNKT cells (Supplementary Fig. 5d). These data implied that ZAP70 catalytic activity is required during the commitment to become an iNKT cell, but perhaps less so during the expansion phase of these cells that occurs later. Importantly, the percentage of iNKT cells that were Tbet^hi in the iNKT cell population in cultures exposed to inhibitor from day 2 did not change compared to DMSO treated controls, while the expression of PLZF was reduced in a dose-dependent manner (Fig. 3e, f). We also consistently observed a higher sensitivity of expression of the NKT2/NKT17 marker PD-1 to chemical inhibition, and in contrast, a higher inhibitor tolerance for expression of the NKT1 marker CXCR3 (Fig. 3g, h), which is under the control of T-bet[39]. Overall, these results demonstrated that PLZF expression and the NKT2 population were more sensitive to diminution of TCR signaling, and conversely, that NKT1 cells can withstand a lower TCR signal strength.

**NKT1 cells are similar to wild type in SKG mice.** We performed RNA-seq to compare the global gene expression pattern of thymic iNKT cell subsets from WT and SKG mice. We defined the subsets by surface expression of several proteins, as described[12,36,40] (Online methods and Supplementary Fig. 6). The expression pattern of signature transcription factors was used to confirm enrichment of each subset. Principal component analysis (PCA) of the RNA-seq data showed three clearly distinct iNKT cell populations (Fig. 4a). Importantly, each subset from SKG mice clustered together with its WT counterpart. In particular, NKT1 thymocytes from SKG mice were in close proximity to WT NKT1 thymocytes in the PCA plot, suggesting that the very predominant NKT1 cell subset in SKG mice was comparable to

WT NKT1 cells. In agreement with this, there were only 158 genes with a greater than 1.5-fold change in expression comparing SKG and WT in NKT1 cells, with 101 of these down in SKG NKT1 cells, whereas much greater numbers of genes distinguished the NKT2 and NKT17 thymocytes from these strains (Supplementary Fig. 7a–c).

**Reduced inhibitory receptors in SKG NKT1 cells.** Pathway enrichment analysis of the differentially expressed genes in NKT1 thymocytes from the two mouse strains revealed that the TCR signaling pathway was significantly affected, with reduced expression of inhibitory receptors and genes that have the potential to inhibit signal transduction in SKG NKT1 cells. For example, there was lower expression of the genes encoding inhibitory surface proteins *Pdcd1*, *Tigit*, *Cd160*, and *Cd5* in SKG NKT1 cells compared to WT NKT1 cells (Fig. 4d). Natural killer (NK) receptors have been shown to be NKT1-specific markers in both BALB/c and C57BL/6[13] mice, and interestingly NK cell-mediated cytotoxicity showed the highest significant *p* value in the pathway analysis (Fig. 4b). As published previously[34,41], the majority of NK receptor genes that were highly transcribed in WT NKT1 cells encode inhibitory receptors, including Klrd1/CD94, Klra5/Ly49e, Klra3/Ly49c, Klra1/Ly49a, Klrc1/NKG2A, and Klra7/Ly49G (Fig. 4c). We found that transcription of most NK receptors was significantly reduced in SKG iNKT1 thymocytes compared to WT counterparts. This is consistent with the requirement for TCR signaling for the upregulation of these receptors[42–44]. WT iNKT cells also expressed the activating NK receptors NKG2D and NKp46, however, *Klrk1* transcripts (NKG2D) did not differ in SKG NKT1 cells (Fig. 4c) and *Ncr1* (NKp46) was transcribed at a low level in both strains.

Genes related to T helper cell differentiation and function, including the IFN-γ and IL-4 signaling pathways and JAK/STAT pathways involved in cytokine signaling were also altered in SKG NKT1 cells (Fig. 4b) as was expression of *Lat2*, *Sprouty*, *Tnfaip3*, and *Socs3* (Fig. 4d). LAT2 is a transmembrane adaptor protein closely related to LAT, but its deletion resulted in hyper-activated T cells and autoimmune disease[24]. Sprouty2 (Spry2) is a negative regulator of receptor tyrosine kinase signaling[45–47]. TNFAIP3 (A20) is a ubiquitin-editing enzyme that suppresses NF-κB activation. Suppressor of cytokine signaling 3 (SOCS3) SOCS3 is capable of inhibiting STAT4, which is activated by IL-12[48], and therefore, can potentially influence NKT1 function. Therefore, compared to WT NKT1 thymocytes, SKG NKT1 cells have reduced expression of inhibitory receptors and molecules involved in the inhibition of signal transduction, perhaps a necessary compensation for the reduced TCR signaling capacity resulting from the SKG *Zap70* mutation.

**Increased NKT1 cells in the spleen of SKG mice.** The difference in iNKT cell subsets in SKG mice was not confined to the thymus. We found a similar difference in TCRβ expression (Supplementary Fig. 8a, b). However, unlike in the thymus, we observed an increase in splenic iNKT cells, both in percentage and cell numbers (Supplementary Fig. 8c, d). Perhaps this expansion is related to the decrease in mainstream CD4 and CD8 T lymphocytes in SKG mice. Importantly, the increase was due to augmented splenic NKT1 cells, and there was no elevation in the relatively minor populations of splenic NKT2 and NKT17 cells (Supplementary Fig. 8e–g). This was reflected by the expression level of signature transcription factors and surface markers (Supplementary Fig. 8h, i). Furthermore, consistent with the thymus RNA-seq data (Fig. 4c, d), splenic SKG iNKT cells showed reduced expression of inhibitory molecules CD160,

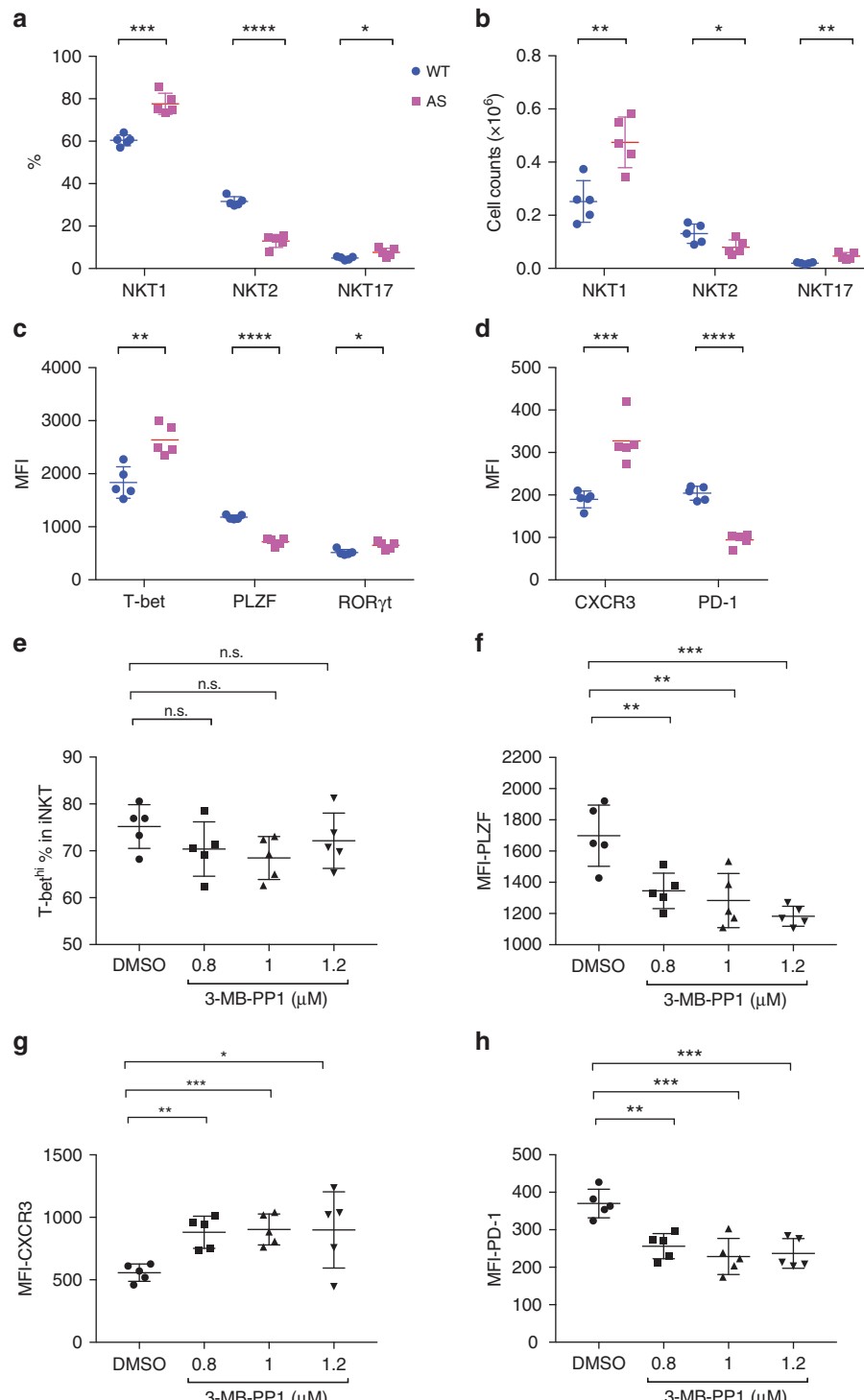

**Fig. 3** NKT1 skewing phenotype in ZAP70AS mice. **a**, **b** Percentages in total iNKT cells and absolute numbers of iNKT cell subsets in ZAP70AS and control mice. **c**, **d** Expression of lineage defining transcription factors and some subset surface markers. **e–h** Day 0 thymic lobes from ZAP70AS mice were cultured as in Fig. 2. Different doses of the inhibitor 3-MB-PP1were added on the second day of culture, and iNKT cells were examined on day 7 for expression of the indicated proteins. Data are representative of three independent experiments. Graphs represent mean ± SD with symbols representing individual mice (**a–d**) or thymic lobes (**e–h**). *$p < 0.05$; **$p < 0.01$; ***$p < 0.001$; ****$p < 0.0001$; n.s. not significant (unpaired two-tailed Student's $t$ test)

NKG2a/c/e, and Ly49G2 when compared to WT iNKT cells (Supplementary Fig. 8j).

**iNKT cells prevent severe arthritis in SKG mice.** Spontaneous arthritis in SKG mice can be accelerated by the injection of the fungal cell-wall component mannan[10]. To determine if iNKT cells have an active role in pathogenesis in SKG mice, we evaluated the presence of iNKT cells in ankles and joint-draining lymph nodes following mannan injection during the course of arthritis. Although the percentage of iNKT cells did not increase due to the influx of other cells, the number of iNKT cells increased in ankles as arthritis progressed, and this number was significantly

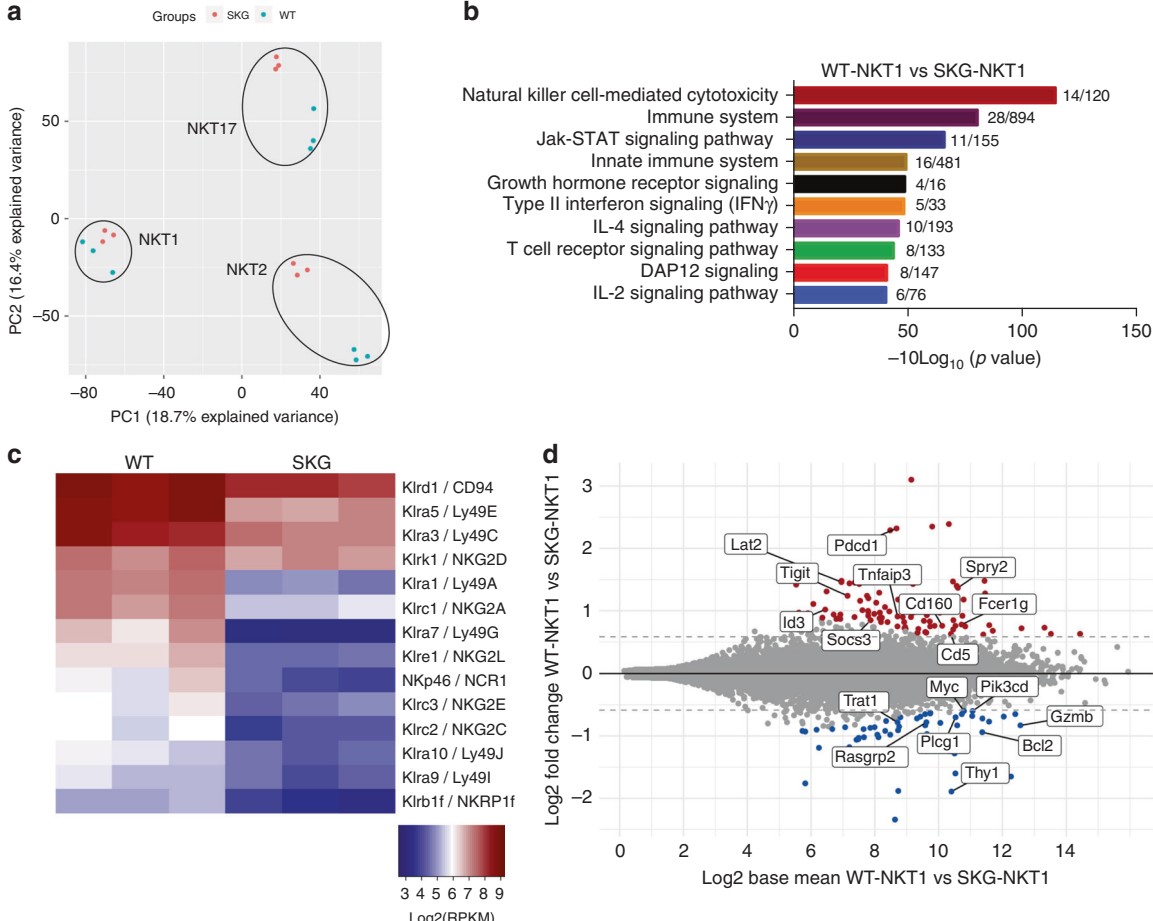

**Fig. 4** Transcriptomic analysis of iNKT cell subsets in WT and SKG mice. **a** Principal component analysis (PCA) of transcriptomes from NKT1, NKT2, and NKT17 cells from WT and SKG thymus tissue. **b** Enriched pathways comparing NKT1 transcriptomes from WT and SKG mice, numbers of genes affected compared to the number in that category are indicated. **c** Unscaled Heatmap showing the expression of NK receptors in NKT1 cells from WT and SKG mice. **d** MA plot of differentially expressed genes (fold change > = 1.5, adjusted $p$ < = 0.1) comparing WT and SKG NKT1 cells

correlated with the disease-associated change in ankle thickness (Fig. 5a, b). Furthermore, expression of the activation marker CD69 was significantly increased on ankle iNKT cells during arthritis development (Fig. 5c). A similar correlation between the number of iNKT cells and the clinical severity of arthritis, as well as an increased expression of CD69 by iNKT cells, was observed in joint-draining lymph nodes (Supplementary Fig. 9a, b). Together, these data suggest iNKT cells might play a significant role in arthritis in SKG mice.

To assess the role of iNKT cells more directly, we evaluated arthritis development in iNKT cell-deficient ($Cd1d^{-/-}$) and iNKT cell sufficient ($Cd1d^{+/-}$) SKG mice. The absence of iNKT cells significantly aggravated the development of arthritis, with regard to clinical signs, histopathological scoring of synovial inflammation, bone destruction and cartilage depletion (Fig. 5d, e). Further visual evaluation of bone morphology by micro-CT analysis confirmed increased signs of bone destruction in iNKT cell-deficient SKG mice (Fig. 5f). These results strongly suggest that iNKT cells were necessary to protect against severe arthritis development in SKG mice.

Deficiency of CD1d in SKG mice could potentially create an empty developmental niche in which other T cells could differentiate and expand, or it could affect CD1d reactive cells with more diverse TCRs. To circumvent this, we acutely deleted iNKT cells from WT SKG mice using NKT14, an antibody that specifically depletes iNKT cells based on reactivity to the

invariant TCR α chain[49]. We verified the efficacy of iNKT cell depletion in joint-draining lymph nodes of the antibody treated mice by flow cytometry (Fig. 5g). Similar to $Cd1d^{-/-}$ SKG mice, antibody depletion of iNKT cells was associated with significantly aggravated arthritis development (Fig. 5h, i). Together, these results show that iNKT cells were required to protect SKG mice from the development of severe arthritis.

To confirm a suppressive effect of iNKT cells on arthritis pathogenesis, we transferred CD4$^+$ SKG T cells alone or in combination with SKG iNKT cells to BALB/c $Rag2^{-/-}$ mice and induced arthritis by mannan injection 1 week after cell transfer (Fig. 5i). Consistent with the known dependence of arthritis on T lymphocytes, transfer of CD4$^+$ SKG T cells promoted arthritis development in $Rag2^{-/-}$ mice (Fig. 5j, k). Importantly, the severity of arthritis was significantly reduced in the presence of SKG iNKT cells, confirming that they have the capacity to ameliorate disease (Fig. 5k).

**Changes of iNKT cell subsets during arthritis development.** The positive correlation between increased joint iNKT cells and arthritis progression led us to investigate if local iNKT cells retained protective function during disease. We hypothesized that different iNKT cell subsets might have distinctive roles as pathogenesis progresses. We therefore assessed the dynamics of iNKT cell subsets in SKG joints and draining lymph nodes at

different stages of arthritis development. Activation of cytokine production by PMA and ionomycin treatment of iNKT cells from arthritic SKG mice revealed that iNKT cells with the capacity to produce IFN-γ (NKT1), IL-4 (NKT2), and IL-17A (NKT17) all were present in joint-draining lymph nodes, with

some IFN-γ plus IL-4 double producers, typical of NKT1 cells. In arthritic joints, however, only iNKT cells capable of producing IFN-γ (NKT1) or IL-17A (NKT17) were present (Fig. 6a). In SKG arthritic ankles, we observed a gradual decline in the frequency of NKT1 cells and an increased frequency of NKT17

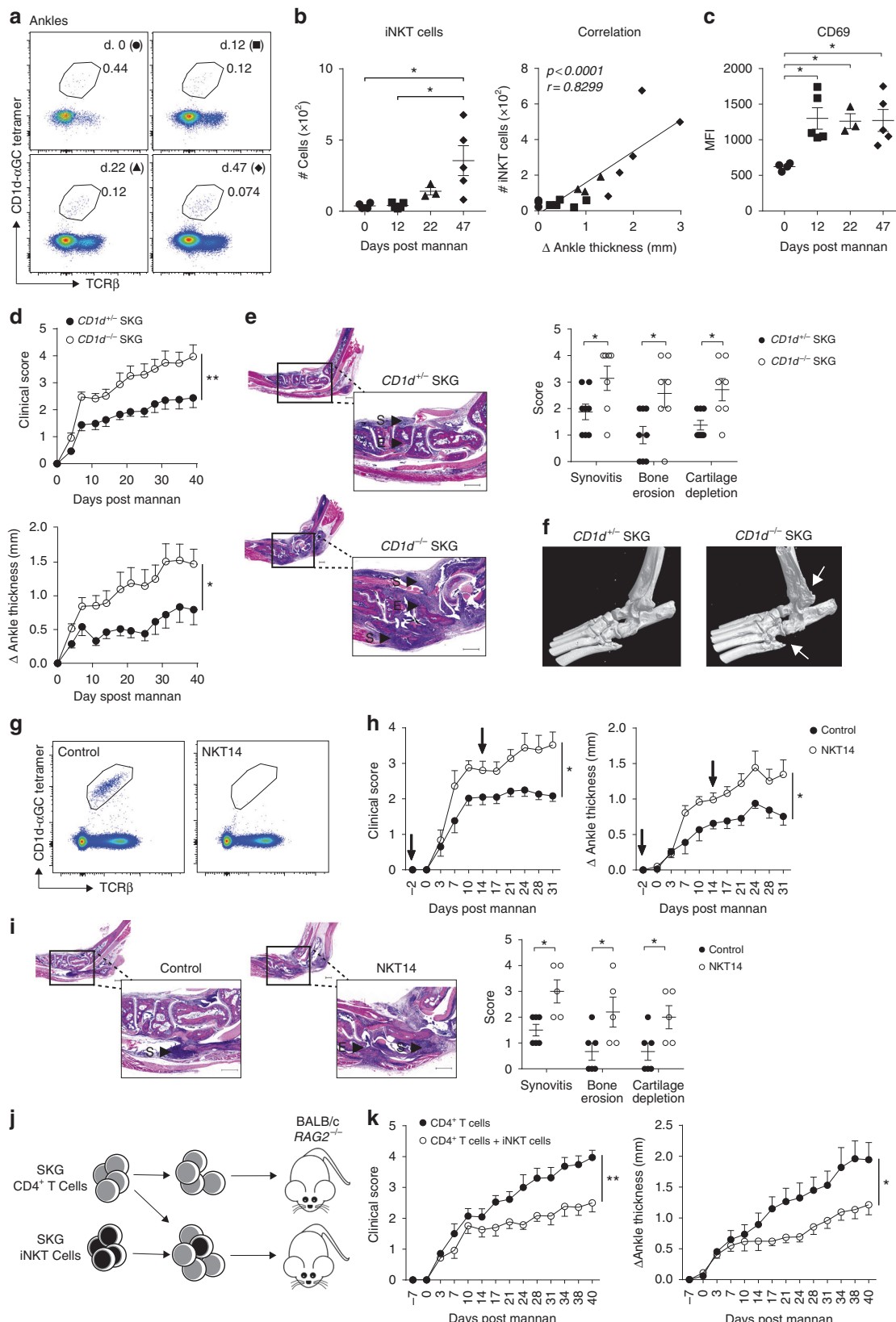

cells during disease progression (Fig. 6b). Importantly, joint NKT1 cell and NKT17 cell frequencies displayed opposite correlations with the severity of arthritis, as measured by changes in ankle swelling (Fig. 6c), with increased NKT17 cells correlating with severity. A progressive increase in NKT17 cell frequency in joint-draining lymph nodes also correlated with the clinical severity of arthritis (Supplementary Fig. 9c–e). However, lymph node NKT1 or NKT2 cell frequencies underwent only minor changes during the course of arthritis that did not correlate with disease severity (Supplementary Fig. 9c–e). Therefore, changes in the frequencies of joint iNKT cell subsets occurred during the course of arthritis. Furthermore, these data suggest that NKT1 cells may mediate the protective activity of iNKT cells in SKG arthritis.

**Impaired IFN-γ from RA synovial fluid iNKT cells.** Previous reports have identified iNKT cells in the synovial fluid (SF) and synovial tissue (ST) of rheumatoid arthritis (RA) patients[50,51]. We confirmed that iNKT cells were present in the SF of patients with RA (Fig. 6e; Supplementary Table 1). Defined functional subsets of iNKT cells, namely NKT1, NKT2, and NKT17, are not as well defined in humans, but we tested the ability of iNKT cells from paired SF and peripheral blood (PB) to produce IFN-γ upon exposure to αGC. IFN-γ production from SF iNKT cells was significantly lower than from matched PB samples (Fig. 6f). Although this could be due to an impaired antigen presentation, stimulation with PMA and ionomycin confirmed a reduced presence of IFN-γ producing iNKT cells in the SF compared to the blood (Fig. 6g). Together with the results obtained from SKG mice, these results are consistent with the hypothesis that active arthritis is correlated with a decreased ability to produce IFN-γ by iNKT cells or reduced NKT1 cells in joint tissue.

**Unimpaired response to cytokines from SKG iNKT cells.** Consistent with the known dampening effect of the $Zap70^{SKG}$ mutation on TCR-signaling, SKG thymic iNKT cells displayed a reduced response to TCR activation in vitro (Supplementary Fig. 3a). Additionally, we found that splenic SKG iNKT cells had reduced production of IFN-γ and IL-4 in response to αGC injection in vivo (Fig. 6h), although the decrease in IFN-γ producing SKG iNKT cells was much less drastic than for IL-4 (Fig. 6h). NKT1 cells also can undergo TCR-independent, cytokine-mediated activation, most efficiently in response to IL-12 plus IL-18 or to LPS[52]. Activated in this way, NKT1 cells produce IFN-γ, but not the mixed, Th0 pattern of IFN-γ plus IL-4, combined with other cytokines, that results from αGC injection into mice. We therefore compared the in vivo iNKT cell response to LPS in WT and SKG mice. LPS promoted an increased

percentage of cells with intracellular IFN-γ from splenic iNKT cells of SKG mice compared to WT mice, likely reflective of the increased number of NKT1 cells (Fig. 6i). This difference was consistent with significantly increased serum levels of IFN-γ 6 h after LPS injection (Fig. 6j). No difference in IFN-γ production from *trans* activated NK cells, meaning NK cells that were activated as a result of iNKT cell stimulation, was observed between SKG and WT mice (Supplementary Fig. 9f). Furthermore, cultures of peripheral lymph node cells revealed that iNKT cells from pre-arthritic or arthritic SKG mice showed an enhanced IL-12-induced expression of IFN-γ by intracellular cytokine staining (Fig. 6k), despite having a less pronounced increase in NKT1 cells compared to thymus or spleen (Supplementary Fig. 9g, h). Together these results suggest that despite having reduced antigen sensitivity of their TCR, SKG iNKT cells showed an increased IFN-γ response to cytokine-mediated activation.

**iNKT cells suppress arthritis through IFNγ.** IFN-γ is protective in several models of arthritis, including the SKG model[53,54]. Due to the enrichment of NKT1 cells in SKG mice, we therefore hypothesized that the arthritis-ameliorating effect provided by iNKT cells could be mediated through the production of IFN-γ. To evaluate this hypothesis, we activated iNKT cells from WT and $Ifng^{-/-}$ (IFN-γ-KO) SKG mice using αGC on the same day as arthritis induction by mannan. Injection with αGC in WT mice caused a significant reduction in arthritis development when compared to PBS treated controls (Fig. 7a). However, αGC injection did not affect arthritis development in IFN-γ-KO SKG mice (Fig. 7b), despite the induction of other cytokines by αGC injection, suggesting that the protective effect provided by iNKT cells required IFN-γ.

Although our results indicate that activation of IFN-γ production by iNKT cells might provide a therapeutic regimen for RA, injection of αGC into SKG mice with established arthritis did not reduce disease (Supplementary Fig. 10a). αGC is a very potent antigen for iNKT cells, but we reasoned that due to its activation of many cytokines, the relative paucity of NKT1 cells in the joints when arthritis is established, or the poor TCR signaling by SKG iNKT cells, that this antigen might result in insufficient IFN-γ production. The glycolipid antigen 7DW8-5, with a *para*-fluoro-phenyl modification of the αGC sphingosine base, has previously been described to elicit an even stronger Th1-biased response than αGC[55]. Treatment with 7DW8-5 on the day of mannan injection also prevented the development of arthritis in SKG mice (Supplementary Fig. 10b). Importantly, therapeutic treatment of arthritic SKG mice with 7DW8-5 significantly reduced the progression of arthritis (Fig. 7c).

**Fig. 5** iNKT cells suppress arthritis. **a** Representative flow cytometry plots of iNKT cells (TCRβ$^{int}$ CD1d-αGC tetramer$^+$ in live CD8$^-$CD19$^-$ cells), in ankles of SKG mice during the course of arthritis. Day 0 (circle), day 12 (square), day 22 (triangle), and day 47 (diamond) post-mannan injection. **b** Number of iNKT cells in ankles (left panel) and correlation with ankle swelling (right panel) during the course of arthritis in SKG mice. **c** Expression of CD69 on ankle iNKT cells during the course of arthritis in SKG mice. **d** Clinical score (left panel) and change in ankle thickness (right panel) in $Cd1d^{-/-}$ ($n = 6$) and $Cd1d^{+/-}$ ($n = 8$) male SKG mice after injection of mannan. **e** Histological evaluation of synovial inflammation (S), bone erosion (E), and cartilage destruction in arthritic ankles from $Cd1d^{-/-}$ ($n = 6$) and $Cd1d^{+/-}$ ($n = 8$) SKG mice. Scale bar 500 μM. **f** Representative Micro-CT of ankles from arthritic $Cd1d^{-/-}$ and $Cd1d^{+/-}$ SKG mice. White arrow indicates bone erosion and reactive bone formation which is increased in $CD1d^{-/-}$ SKG mice. **g** Representative flow cytometry plots of iNKT (gated as in **a**) cell depletion efficacy in lymph nodes after injection with iNKT cell-depleting (NKT14) mAb. **h** Clinical score (left panel) and change in ankle thickness (right panel) in male SKG mice treated with NKT14 mAb ($n = 5$) or isotype control ($n = 6$) after injection of mannan. Arrows indicate time of injection. **i** Histological evaluation of synovial inflammation (S), bone erosion (E), and cartilage depletion in male SKG mice treated with control or NKT14 mAb. Scale bar 500 μM. **j**, **k** Transfer of CD4$^+$ SKG T cells ($1 \times 10^6$) alone ($n = 7$) or in combination with SKG iNKT cells ($3 \times 10^5$) to $Rag2^{-/-}$ mice ($n = 7$) and the clinical score (**k**, left panel) and change in ankle thickness (**k**, right panel) after injection of mannan. Representative data from two independent experiments shown in **a–c**. Complied data from at least two independent experiments are shown in **d**, **e**, **h**, **i**, **k**. Graphs represent mean ± SEM with symbols representing individual mice. *$p < 0.05$, **$p < 0.01$ (Mann–Whitney on area under the curve (**d**, **h**, **k**), Mann–Whitney (**e**, **i**), One way ANOVA (**b**, **c**), Spearman correlation (**b**)

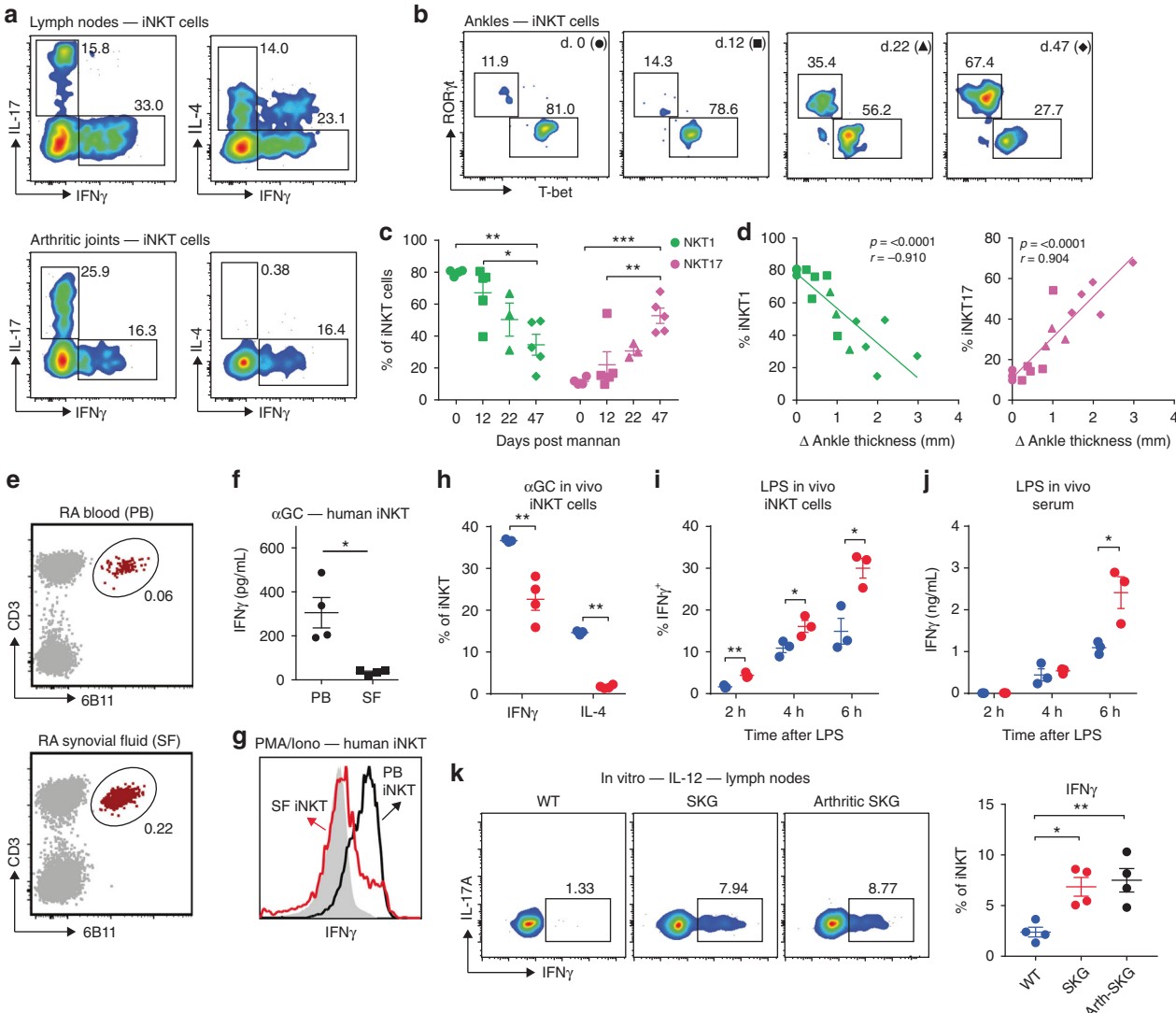

**Fig. 6** NKT1 cells negatively correlate with arthritis severity in SKG mice. **a** Representative flow cytometry plot of cytokine profile of iNKT (TCRβ$^{int}$ CD1d-αGC tetramer$^+$) cells from joint-draining lymph nodes and joints (wrists and ankles) of arthritic SKG mice stimulated with PMA and ionomycin. **b** Representative flow cytometry of iNKT (TCRβ$^{int}$ CD1d-αGC tetramer$^+$) cell subsets in ankles during the course of mannan-induced arthritis in male SKG mice. Day 0 (circle), day 12 (square), day 22 (triangle), and day 47 (diamond) post-mannan injection. **c** Frequency of NKT1 (T-bet$^+$RORγt$^-$, left, green) and NKT17 cells (T-bet$^-$RORγt$^+$, right, purple) during the course of mannan-induced arthritis in SKG mice. **d** Correlation between NKT1 (left panel, green) and NKT17 (right panel, purple) and changes in ankle swelling in mice with mannan-induced arthritis. **e** Representative flow cytometry plot of iNKT cells (CD3$^+$Vα24Jα18$^+$ (6B11)) in paired peripheral blood (PB) and synovial fluid (SF) samples derived from rheumatoid arthritis (RA) patients ($n = 8$). **f** Levels of IFNγ in paired peripheral blood (PB) and synovial fluid (SF) samples stimulated with αGC (RA patients, $n = 4$). **g** Representative histogram (of a total of 5 RA patients) of IFN-γ expressing iNKT cells (CD3$^+$Vα24Jα18$^+$ (6B11 antibody-reactive) after stimulation with PMA and ionomycin (black/red histogram) or unstimulated control (filled histogram). **h** Expression of IFN-γ and IL-4 from splenic iNKT cells after injection of αGC in WT BALB/c (blue) or SKG mice (red). **i** Expression of IFN-γ by splenic iNKT cells after injection of LPS in WT BALB/c (blue) or SKG mice (red). **j** Serum IFNγ in WT BALB/c (blue) or SKG mice (red) after LPS injection. **k** Production of IFN-γ by iNKT cells after stimulation of lymph node cultures for 48 h with IL-12 (20 ng/ml). Lymph node cultures prepared from WT BALB/c (blue), pre-arthritic SKG (red, 7–8 weeks old) and arthritic SKG mice (black, 40 days post-mannan injection). Representative of at least two independent experiments are shown. Graphs represent mean ± SEM with symbols representing individual mice. *$p < 0.05$, **$p < 0.01$ ***$p < 0.001$. (One way ANOVA (**c**,**k**), paired two-tailed $t$ test (**f**), unpaired two-tailed Student's $t$ test (**h**–**j**), Spearman correlation test (**d**))

## Discussion

We have shown that decreased TCR signal strength due to Zap70 mutations affected the number of thymus iNKT cells, and more strikingly, led to a predominance of NKT1 cells. Furthermore, we demonstrated that the IFN-γ induced by iNKT cell activation protected from severe arthritis in ZAP70 mutant mice, although the representation of potentially protective NKT1 cells diminished in the ankles as disease progressed. Therefore, our data show that a central immune defect, an alteration in thymic selection due to reduced TCR signal strength, led not only to arthritogenic Th17 cells, as reported previously[9,10], but also to an increased NKT1 population that ameliorated the most severe form of disease.

On account of their limited TCR repertoire, it is surprising that thymic iNKT cells can differentiate into functional subsets. Differences in TCR signal strength could be an influence on iNKT

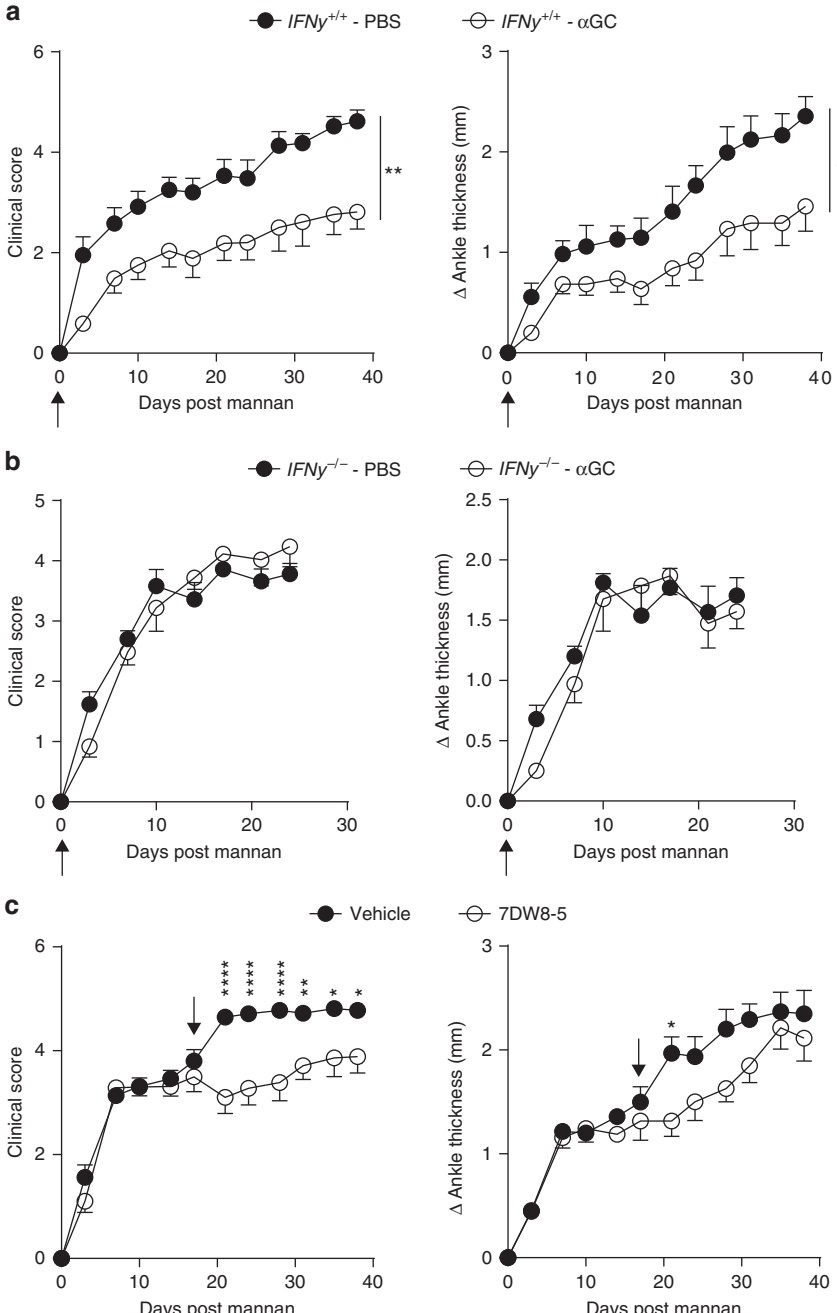

**Fig. 7** NKT1-biased activation suppresses progression of arthritis in SKG mice. **a** Clinical score (left panel) and change in ankle thickness (right panel) in WT (*Ifng*[+/+]) SKG mice with mannan-induced arthritis after treatment with either PBS (r.o., $n = 6$) or αGC (4 μg, r.o., $n = 8$) at the day of arthritis induction (indicated by arrow). (**b**) Clinical score (left panel) and change in ankle thickness (right panel) in IFN-γ-KO (*Ifng*[−/−]) SKG mice with mannan-induced arthritis after treatment with either PBS (r.o., $n = 5$) or αGC (4 μg, r.o., $n = 6$). αGC was injected as above. **c** Clinical score (left panel) and change in ankle thickness (right panel) in SKG mice with mannan-induced arthritis treated with either vehicle control (i.p. $n = 8$) or 7DW8-5 (4 μg, i.p., $n = 8$) 17 days after mannan injection (indicated by arrow). Compiled data from at least three independent experiments are shown. Graphs represents mean ± SEM. *$p < 0.05$, **$p < 0.01$ ***$p < 0.001$, ****$p < 0.0001$. (Mann–Whitney on the area under curve (**a**, **b**), Two-way ANOVA (**c**))

cell subset formation, however, considering the diversity of iNKT cell β chains and other factors that might affect TCR avidity for selecting ligands. Here we have analyzed the broad iNKT cell population and show that TCR signal strength does influence iNKT cell subsets. Although iNKT cells are agonist selected, NKT1 cells were favored when TCR signaling was lessened due to a hypo-functional ZAP70 protein. Moreover, using FTOC, we have shown that catalytic inhibition of ZAP70 after culture initiation reduced the high expression of PLZF and characteristic

NKT2 cell surface proteins in iNKT cells, whereas expression of proteins characteristic of NKT1 cells was spared. These results indicated that continuing strong TCR signals were required for the maintenance of high amounts of PLZF, whereas NKT1 cell differentiation withstood the lower TCR signals. Given that Egr2 can directly bind to the *Il2rb* locus encoding CD122[16], we speculate that the TCR signal strength in SKG mice was sufficient to increase the required amount of CD122 in differentiating and expanding populations of NKT1 cells, but not enough to

maintain the high expression of PLZF critical for NKT2 and NKT17 cell development. We note that TCR signal strength is not the only influence on iNKT cell subsets, and it is well established that cytokines are also important, such as the IL-17RB subunit of the IL-25 receptor for NKT2 cells and IL-15 signaling for NKT1 cells.

Our conclusion regarding the reduced requirement for TCR signal strength for NKT1 cells is consistent with a recent study that used retrogenic mice to assess the effect of expression of a single TCRβ chain on iNKT cell subsets. Expression of a TCRβ chain that imparted an increased TCR half-life for binding to αGC tetramers and increased conjugate formation with CD1d[+] thymocytes, correlated with an increase in NKT2 cells[56]. However, only a few engineered TCRs were sampled and the in vitro binding assays may not reflect the in vivo interactions between endogenous ligands and iNKT cell precursors. Another group reported that expression of CD5 and the Nurr77 reporter for TCR activation both were increased in NKT2 cells[36], but the data did not distinguish if this was a requirement for the NKT2 cell population, or a result of their tonic stimulation after differentiation. We also detected a higher $Nur77^{gfp}$ reporter signal from NKT2-like cells in FTOC compared to NKT1-like cells, however, stage 0 iNKT cells had a much higher reporter signal. These data suggest that the highest TCR signals were achieved during the initial positive selection of iNKT cells, but also are consistent with our evidence that stronger signals are required by NKT2 as opposed to NKT1 cells throughout their differentiation and expansion.

We found that iNKT cells from SKG mice protected from severe, mannan-accelerated disease, and that the protection imparted by prophylactic αGC injection required IFNγ. Additionally, the 7DW8-5 antigen, previously reported to induce an even more IFNγ-skewed response than αGC, exhibited therapeutic potency, in contrast to αGC. Arthritis in SKG mice is due to autoreactive T cells that produce IL-17, and therefore, our data are consistent with other findings indicating that high levels of IFN-γ can suppress IL-17 secretion and that IFN-γ is protective in arthritis models[53,54]. The protective IFN-γ induced by iNKT cell stimulation need not be entirely from iNKT cells, as NK cells and other cells can be induced to produce IFN-γ downstream of iNKT cell activation.

Despite their ability to prevent severe arthritis, iNKT cells from SKG mice were less responsive than WT cells to antigenic stimulation. The transcriptomic analysis indicated that SKG NKT1 cells may have adapted during differentiation to decreased TCR signaling, including a reduction in the expression of inhibitory receptors and molecules transducing inhibitory signals. The decrease in inhibitory signaling may be responsible for the increased IFN-γ response SKG iNKT cells show to TCR-independent inflammatory signals, such as exposure to LPS or IL-12. The adaptation of SKG thymic iNKT cells to reduced TCR signals is indicative of the broad effects on T cell selection in SKG mice, including changes in the TCR repertoire due to increased secondary rearrangements to distal $Traj$ segments. Therefore, we propose that the increased number of NKT1 cells in SKG mice, their reduction in inhibitory receptors, and their enhanced response to inflammatory cytokine signals, may have allowed them to compensate for their partially impaired TCR signaling in protecting these mice from severe arthritis.

In previous studies using collagen-induced arthritis (CIA), iNKT-deficient mice developed ameliorated rather than enhanced disease[57]. It is known that CIA relies on immunization and generation of antigen-specific antibodies for disease pathogenesis. iNKT cells are a significant inducer of humoral responses in several contexts[58,59], therefore, the reduced development of CIA in iNKT-deficient mice could be due to an impaired induction of autoantibody production. In adjuvant induced arthritis, however, regulatory B cell–iNKT cell interactions ameliorated disease through stimulation of IFN-γ[60]. The etiology and immunopathological mechanism are entirely different in SKG mice, in which autoimmune T cells resulting from defective thymic selection drive disease[9]. Different mouse models of RA, including CIA and SKG, may recapitulate different aspects of RA pathogenesis that can variably dominate disease development among patients[61]. Thus, the above-mentioned differences between CIA and SKG also suggest that the protective role of iNKT cells in RA might be more prominent in specific patient subsets.

In the absence of immunization with exogenous antigen, the activity of iNKT cells did not completely prevent arthritis. As disease progressed, Th17-driven immune responses increased in SKG mice and NKT17 cells became more prevalent in the ankle at the expense of NKT1 cells. It is not known if this dynamic change is caused by cell recruitment or the ability of NKT1 cells to become NKT17 cells, nor is it known if the NKT17 cells make an important contribution to pathogenesis. Regardless, this change clearly reflects a diminution of iNKT cell protective capacity. We observed a capacity for IFN-γ production from iNKT cells from the synovial fluid of RA patients, but interestingly, this was reduced compared to iNKT cells from PB of the same patients. Large scale studies will be required, however, to determine if the prevalence of iNKT cells capable of producing IFN-γ in synovial fluid or joint tissue correlates with disease severity in patients. Nevertheless, the ability of circulating iNKT cells from patients to produce this cytokine, and the therapeutic effect of an antigen inducing the highest amount of IFN-γ in mice, suggest that boosting IFN-γ from iNKT cells, or from other innate-like T cells such as MAIT cells, could have therapeutic benefit.

In conclusion, our study highlights how thymic maturation of T cells influences the effector functions of various T cell populations, including innate-like T cells, which orchestrate complex immune responses. The study also shows how the severity of an inflammatory disease process can be influenced by thymic selection. Therefore, understanding the mechanism of T cell differentiation could have important implications for designing therapeutic interventions in human disease.

## Methods

**Mice**. SKG mice were obtained from S. Sakaguchi (Osaka University). For comparisons between WT and SKG, BALB/cJ mice from Jackson Laboratories (Jax 000651) were crossed with SKG mice to generate an F1. WT and SKG from F2 offspring were bred separately to generate WT and SKG F3, which were used in this study. ZAP70AS mice were obtained from A. Weiss (UCSF), and the WT controls and AS mice used in this study were the F3 offspring from backcrossing ZAP70AS to C57BL/6J (JAX 000664) as described above. BALB/c $Cd1d^{-/-}$ (C.129S2-$Cd1^{tm1Gru}$/J), BALB/c CD45.1 (CByJ.SJL(B6)-$Ptprc^a$/J), and BALB/c $Ifng^{-/-}$ (C.129S7(B6)-$Ifng^{tm1Ts}$/J) mice were all obtained from Jackson Laboratories. BALB/c $Rag2^{-/-}$ mice were purchased from Taconic. Mice were bred and housed at the La Jolla Institute for Allergy and Immunology (LJI) vivarium and the Altman Clinical and Translational Research Institute vivarium (ACTRI) at UCSD under specific pathogen free conditions (SPF). All experiments were conducted in accordance with the protocol approved by the Institutional Animal Care and Use Committees of LJI and UCSD.

**Flow cytometry and antibodies**. Single-cell suspensions were prepared from thymus, spleens and lymph nodes. For isolation of synovial cells, joints (wrist and/ or ankles) were collected, the tibia and digits removed and remaining pieces rinsed with RPMI1640 (Corning) to avoid bone marrow contamination. Joints were dissociated with Liberase TM (Roche) at 37 °C. Before antibody staining cells were pre-incubated with Fc block (BD Biosciences). CD1d-αGC tetramers were prepared in our laboratory as described previously[62] and used at a dilution of 1:200. Staining for transcription factors was performed using reagents and protocols from the Transcription Factor (TF) Buffer Set (BD Biosciences). For intracellular cytokine staining, Cell Fixation/Permeabilization kit (BD Biosciences) was used. Proliferation of DP cells was examined using BD PharMingen BrdU Flow Kits. Briefly, 0.7 mg of BrdU solution (10 mg/ml) was injected i.p. into each mouse, and 20 h later, incorporation in thymocytes was examined following manufacturer's protocol.

Apoptosis of DP cells was examined using APC Annexin V (BD PharMingen) and supporting protocols. The complete list of other antibodies and reagents used is as follows: Live/Dead-Yellow (ThermoFisher), Fixable viability dye (ThermoFisher), anti-TCRβ (H57-597), anti-CD4 (RM4-5), anti-CD8α (53–6.7), anti-CD19 (6D5), anti-CD24 (M1/69), anti-CD44 (IM7) and anti-NK1.1 (PK136), anti-CXCR3 (CXCR3-173), anti-CD49A (Ha31/8), anti-CD122(TM-beta1), anti-CD5(53–7.3), anti-PD-1(29F.1A12), anti-ICOS (C398.4A), anti-CD1d(1B1), anti-CD69(H1.2F3), anti-Sdc-1(281–2), anti-CD160(eBioCNX46-3), anti-Tigit(GIGD7), anti-NKG2A/C/E(20d5) anti-IFN-γ (XMG1.2), anti-IL-4 (BVD6-24G2), anti-IL-17A (TC11-18H10.1 and eBio17B17), anti-PLZF (R17-809), anti-T-bet (O4-46), anti-RORγt (Q31-378 or B2D), anti-Eomes (Dan11mag), anti-Egr2 (erongr2), anti-Ly49G2 (4D11). Stained samples were analyzed using either LSRII or Fortessa flow cytometers (BD Biosciences) or a ZE5 (Bio-Rad) flow cytometer and FlowJo software (Treestar).

**In vitro and in vivo cytokine production**. For in vitro αGC stimulation, purified CD1d protein was used to coat microwells at 10 μg/ml in PBS at 37 °C for 1 h, then the plate was washed with T cell culture medium TCM (RPMI medium supplemented with 10% (vol/vol) FBS, 50 μM β-mercaptoethanol, 50 μg/ml penicillin/streptomycin/glutamine mix, 10 mM Hepes, 1× MEM nonessential amino acids, 1 mM sodium pyruvate). Thymocytes were enriched for iNKT cells by negative selection of CD8β+ CD62L+ cells. The remaining cells were suspended at $10^8$/ml, incubated with 1 μg/ml of Streptavidin A (Sigma-Aldrich) and washed once with TCM. Enriched thymocytes were added to the CD1d coated wells at $2 \times 10^6$/ml with αGC at 0.5 μg/ml. Cells were incubated for 4 h at 37 °C with Brefeldin A (Sigma-Aldrich) added for the final 2 h. For in vitro stimulation with PMA and ionomycin, thymocytes were enriched as above and incubated with PMA (phorbol 12-myristate 13-acetate; 50 ng/ml) and ionomycin (1.5 μM) for 4 h with Brefeldin A (Sigma-Aldrich) added for the final 2 h, and intracellular cytokines were analyzed by flow cytometry. For in vivo stimulation of iNKT cells, mice were injected i.v. with 1 μg αGC (α-galactosylceramide or KRN7000, Kirin Hakko Kirin California) and spleens were analyzed 1.5 h later. For in vivo LPS challenge, mice were injected i.v. with 50 μg lipopolysaccharide (LPS) from *Salmonella enterica* (Sigma-Aldrich) and spleens were analyzed at 2, 4, and 6 h after injection.

Synovial cells and cells isolated from joint-draining lymph nodes (brachial, axillary and popliteal) from arthritic SKG mice were activated in vitro with PMA and ionomycin as described above.

For in vitro IL-12 stimulation, single-cell suspensions were prepared from pooled lymph nodes (brachial, axillary, inguinal, and popliteal) from WT, SKG, and SKG mice with mannan-induced arthritis. Cells ($5 \times 10^6$/ml) were stimulated for 48 h with 20 ng/ml recombinant mouse IL-12 (R&D Systems) and Brefeldin A was added for the last 7 h of stimulation.

**Mixed bone marrow chimeras**. Total bone marrow cells were prepared from the femurs and tibias of wild-type BALB/c (CD45.1+) or SKG (CD45.2+) donor mice, and samples were depleted of mature T cells with anti-Thy1.2 (30-H12; BD). Recipient F1 mice (CD45.1+CD45.2+) were pretreated with sulfatrim in drinking water 3 days prior, lethally irradiated (900 rads) and received $10 \times 10^6$ T cell-depleted bone marrow cells. Chimeras were maintained on drinking water with sulfatrim and analyzed 6 weeks after transplantation.

**Calcium flux assay**. As described[63], thymocytes from WT mice were labeled with CFSE (20 nM) in PBS for 10' at 37 °C, washed and mixed with thymocytes from SKG mice(CSFE−) at 1:1 ratio. Cells were then loaded with Indo-1 AM (ThermoFisher) 2 uM with 4 μM probenecid (ThermoFisher) in RPMI with 1% FBS. After washing, cells were stained with antibodies against CD4 and CD8α as well as biotinylated anti-CD3ε antibodies (10 μg/ml, 145-2C11; Biolegend) on ice. The cells were stimulated by streptavidin crosslinking in PBS and calcium flux was initiated after addition of CaCl₂(5 mM). The mean fluorescence ratio of Indo-1 violet to Indo-1 blue was calculated using FlowJo software (Tree Star, Inc).

**Thymic organ culture**. Timed breedings were set up and thymic lobes were harvested from e18.5 embryos or d0 mice. Lobes were cultured on cell culture inserts with a 0.4 μm pore in 6-well culture dishes (Sigma #CLS3450) on top of 1.5 ml complete DMEM supplemented with 10% FBS, 50 μM β-mercaptoethanol, 50 μg/ml penicillin/streptomycin/glutamine mix, 10 mM Hepes, 1× MEM nonessential amino acids and 1 mM sodium pyruvate. Media containing DMSO or various concentrations of 3-MB-PP1 was exchanged daily until thymic lobes were dissociated into single-cell suspensions for staining with antibodies for flow cytometric analysis.

**Compounds**. 3-MB-PP1 (3-methylbenzyl-pyrazolopyrimidine) was purchased from Millipore (#529582). 7DW8-5 [(2S,3S,4R)-1-O-(α-D-galactopyranosyl)-N-(11-(4-fluorophenyl)undecanoyl)-2-amino-1,3,4-octadecanetriol] was purchased from Diagnocine (#7DW8-5).

**Isolation of iNKT cell subsets and pre-selection DP cells**. For RNA isolation of thymic iNKT cell subsets, as described previously[12], thymocytes from WT and SKG

mice were enriched for iNKT cells and stained with Live/dead-yellow dye and antibodies against other surface markers. iNKT subsets were sorted using a FACSAria Fusion (BD Biosciences) as in Supplementary Fig. 6: NKT1, CD1d-αGCtetramer+ICOS$^{lo}$CD49a+; NKT2, CD1d-αGCtetramer+CD49−Sdc-1−ICOS$^{h-i}$CD4$^{hi}$TCRβ$^{hi}$; NKT17, CD1d-αGCtetramer+CD49−ICOS$^{hi}$Sdc-1+. Pre-selection DP cells were sorted as CD4+CD8α+TCRβ−CD69−CD5− from un-enriched thymocytes.

**RNA-sequencing**. Sorted cells were lysed in Trizol LS (ThermoFisher), RNA was prepared using miRNeasy micro kit from Qiagen and quantified as described previously[64]. Adapted from the Smart-seq2 protocol[65], an oligo-based capture of Poly-A tailed mRNA was carried out followed by a high-fidelity reverse transcription using locked nucleic acids (LNA)-oligos to obtain full-length transcripts. mRNA complementary (c)DNA strands were then amplified using a PCR-based method and sequencing adaptors incorporated into the cDNA fragments in a single step using Tn5 transposase technology (Nextera XT reagents and barcoded adaptors kits). Libraries were sequenced on the HiSeq2500 Illumina platform to obtain 50-bp single-end reads (TruSeq® Rapid Kit, Illumina).

**RNA-seq analysis**. The single-end reads that passed Illumina filters were filtered for reads aligning to tRNA, rRNA, adapter sequences, and spike-in controls. The reads were then aligned to UCSC mm9 reference genome using TopHat (v 1.4.1). DUST scores were calculated with PRINSEQ Lite (v 0.20.3) and low-complexity reads (DUST > 4) were removed from the BAM files. The alignment results were parsed via the SAMtools to generate SAM files. Read counts to each genomic feature were obtained with the htseq-count program (v 0.6.0) using the "union" option. After removing absent features (zero counts in all samples), the raw counts were then imported to R/Bioconductor package DESeq2 to identify differentially expressed genes among samples. DESeq2 normalizes counts by dividing each column of the count table (samples) by the size factor of this column. The size factor is calculated by dividing the samples by geometric means of the genes. This brings the count values to a common scale suitable for comparison. *p*-values for differential expression are calculated using binomial test for differences between the base means of two conditions. These *p*-values are then adjusted for multiple test correction using the Benjamini Hochberg algorithm to control the false discovery rate. We considered genes differentially expressed between two groups of samples when the DESeq2 analysis resulted in an adjusted *p*-value of <0.1 and the fold change in gene expression was 1.5-fold.

**Gene set enrichment analysis**. The TCR pathway-related gene set was converted to .gmx format. Log to the base 10 of *p* values for all genes were calculated and the sign was changed when there was a negative logfoldchange. These data were used to create the .rnk file. Both gene_sets.gmx and DE.rnk files were loaded in the GSEA GUI and the following parameters were set before running GSEA on a Pre-Ranked gene list (other parameters have default settings): Collapse data set to gene symbols: set as false; Enrichment statistic: classic

**Heatmaps, PCA, and MA plots**. A row-scaled heatmap of log-transformed RPKM values was created with the heatmap.2 function in the gplots v3.0.1 library under R v3.4.2. Hierarchical clustering of samples was done with the 'hclust' function using average linkage. In order to show the expression level of NK receptors in iNKT cells (Fig. 4c), an unscaled heatmap of log-transformed RPKM values was created as above. The genes were arranged in a decreasing order of the average expression level in WT NKT1 samples. PCA was performed using the 'prcomp' function in R, setting the 'scale' parameter to TRUE. (Shiny PCA Maker: https://github.com/LJI-Bioinformatics/Shiny-PCA-Maker). The MA plots were created with ggplot function in the package ggplot2 v2.2.1 under R v3.4.2. We plotted log2 of baseMean and shrunken log2foldchange (both calculated via DESeq2) on the x and y axis, respectively. A fold change cutoff of 1.5 and p-value cut off of 0.1 was applied to color code differentially expressed genes on the plot. The gene names were added using function geom_text_repel in the library ggrepel v0.7.0.

**TCR sequencing**. Pre-selection DP cells from WT or SKG mice were sorted and RNA was extracted as above. cDNA was prepared from RNA with SuperScript III (ThermoFisher). PCR was performed for amplification of the *Trav9*, *Trav11*, and *Trav19* gene segments with specific forward primers for each Vα region (each with an Illumina adaptor sequence on the 5' end: *Trav9*, GCTGCAGCTGCTCCT-CAAGT; *Trav11*, AACAGGACACAGGCAAAG; *Trav19*, GCTGACTGTTCAA-GAGGGA) and a reverse primer for the α-chain constant region (also with an Illumina adaptor sequence on the 5' end; GCACATTGATTTGGGAGTC). PCR products were purified and sequenced on a Miseq system with a MiSeq Reagent Kit v2-500 (Illumina). The paired-end FASTQ files were run through MIXCR v1.8.3 to get clonal sequences which were then primer trimmed using cutadapt version 1.13. These sequences were de-duplicated by removing exact matches. IMGT/HighV-QUEST v1.5.6 was used with default options to determine V and J regions and their usage in the clonal sequences. Libraries greater than 500,000 sequences were split up and run through HighV-Quest separately. The file named "3_Nt-sequences.txt" was used from the output generated by HighV-Quest to process the results.

**Arthritis models**. For mannan-induced arthritis in SKG mice, male and female mice were injected i. p. with 20 mg of mannan (Sigma-Aldrich), dissolved in sterile PBS at 8 weeks of age. Clinical scoring and measurement of ankle thickness using a digital caliper was performed twice weekly according to an established protocol[9]. Briefly, clinical signs of arthritis in front and hind paws were scored as follows: 0, no joint swelling; 0.1 per swollen finger joint (3 digits on front paw and 4 digits on hind paw); 0.5, mild swelling of wrist or ankle; 1.0, severe swelling of wrist or ankle. Scores for all fingers of forepaws and hind paws, wrists and ankles were combined for each mouse yielding a maximum score of 5.4, which was considered the clinical endpoint. All arthritis studies were performed on littermate mice. Mice reaching clinical endpoint scores were sacrificed according to ethical guidelines. Treatment regime and genotypes of mice were blinded to the researcher performing the clinical scoring of arthritis.

For treatment with αGC, mice were injected either on the day of mannan injection (day 0) by retro-orbital injection (r.o, 100 μl) or after establishment of arthritis (day 17) by intraperitoneal injection (i.p, 100 μl). αGC was reconstituted in dH20 to 200 μg/ml and diluted in PBS to 40 μg/ml before injection. PBS was used as control for αGC. For treatment with 7DW8-5 (Funakoshi Co. Ltd, Japan) mice were injected either on the day of mannan injection (day 0) by i.p injection (100 μl) or after establishment of arthritis (day 17) by i.p. injection (100 μl). 7DW8-5 was reconstituted in DMSO to 25 mg/ml and diluted in PBS to 40 μg/ml before injection (final concentration of DMSO <0.2%). PBS with equal concentration of DMSO (Vehicle) was used as control for 7DW8-5.

Depletion of iNKT cells in WT SKG mice was achieved through injection with anti-mouse monoclonal NKT14 antibody (200 μg) obtained from NKT Therapeutics. Mouse IgG2a isotype control (BioXcell) was used as recommended by NKT Therapeutics. The NKT14 antibody has previously been shown to selectively deplete iNKT cells[49,66]. iNKT cells were depleted 2 days before injection of mannan (day −2) and repeated at day 14 after mannan injection. Depletion of iNKT cells were verified by flow cytometric identification of TCRβ+CD1d-αGC tetramer+ cells in joint-draining lymph nodes.

**iNKT and CD4 T cell transfer to RAG2-KO mice**. CD4 T cells were sorted by flow cytometry from the spleen and lymph nodes of female CD45.1 SKG mice and iNKT cells were sorted from the thymus of CD45.2 SKG mice. CD4 T cells ($1 \times 10^6$) were transferred either alone or in combination with iNKT cells ($3 \times 10^5$) to $Rag2^{-/-}$ mice by r.o injection. One week after transfer mice were injected with 20 mg of mannan to induce development of arthritis. Clinical scoring of arthritis was performed in a blinded manner as described above.

**Histological assessment of arthritic joints**. Hind paws were fixed in 10% neutral-buffered formalin, decalcified, and embedded in paraffin. Sections were prepared from tissue blocks by the LJI histology facility and stained with H&E and safranin-O/Fast Green/Hematoxylin (Sigma-Aldrich). Histopathological scoring for inflammation, bone erosion, and cartilage erosion was performed in a blinded manner as previously described[67].

**Micro-CT**. Ankles were place in 10% neutral-buffered formalin. After fixation, samples were transferred to 70% ethanol. Before scanning, bones were transferred to PBS for 48 h. Scanning was performed on a Skyscan1176 micro-CT (Bruker, Antwerp, Belgium) with a voxel size of 9 μm, at 50 kV/ 200 mA, with a 0.5 mm aluminum filter. Exposure time was 810 ms. The X-ray projections were obtained at 0.4° intervals with a scanning angular rotation of 180° and a combination of 4 average frames. The projection images were reconstructed into 3D images using NRECON software (Bruker) and CT-Analyzer (Bruker). Images were generated using the CT-VOX software (Bruker).

**Human iNKT cells**. Paired synovial fluid (SF) and peripheral blood (PB) samples were obtained from eight patients with definitive rheumatoid arthritis (RA), fulfilling the EULAR/ACR2010 criteria (Supplementary Table 1). This study was approved by the local ethics committee of Ghent University Hospital (Belgium) and all patients gave written informed consent. Blood and SF samples were diluted in RPMI medium and mononuclear cells (MC) were isolated by density gradient centrifugation using histopaque®-1077 (Sigma-Aldrich). iNKT cells were defined as live CD3+CD19− TCRVα24Jα18+TCRVβ11+ cells detected within the lymphocyte gate of PBMC and SFMC samples using the L/D stain 7-AAD (BD) and the following monoclonal antibodies: anti-human CD3 APC-eFluor780 (UCHT1), CD19 PERCP-Cy5.5 (HIB19), TCRVα24Jα18 PE (6B11, all from eBiosciences) and TCRVβ11 FITC (C21, Beckman Coulter). Samples were acquired on a FACS Canto II flow cytometer (BD) and analyzed by means of FlowJo 10.3 software (Treestar).

For evaluation of IFN-γ production by iNKT cells directly ex vivo, freshly isolated PBMC and SFMC were incubated for 4 h with phorbol 12-myristate 13-acetate (PMA, 25 ng/ml), calcium-ionomycin (CaI, 1 μg/ml; both from Sigma) and brefeldin A (BFA, 10 ng/ml; BD) or BFA alone (negative control). Cells were prepared for intracellular cytokine staining as described before[68] using the Cytofix/Cytoperm kit (BD), L/D Fixable Violet Dead Cell Stain Kit (ThermoFisher) and anti-human IFN-γ PERCP-Cy5.5 mAb (45.B3, eBioscience).

In separate experiments, $3 \times 10^6$ PBMC and SFMC were incubated for 48 h with αGC (100 ng/ml; Laboratory for Medicinal Chemistry, Ghent University) and IFN-γ levels were measured in supernatants of PB and SF cell cultures by means of ELISA (eBioscience).

**Statistics**. Sample sizes were selected based upon our experience with the above-mentioned assays in order to achieve sufficient power to detect biologically relevant differences in the experiments being conducted with an α error (two tailed) <0.05.

For statistical analysis, a 2-tailed Mann–Whitney test was performed on non-parametric data. On normal distributed data 2-tailed unpaired $t$ test, one way ANOVA or paired $t$ test were performed as reported in the figure legends. Spearman test was used for calculation of correlations. All statistical analyses were performed using GraphPad Prism software. A comparison was considered significant if $p < 0.05$.

**Data availability**. RNA-seq data in this study have been deposited in GEO with the accession codes GSE114555 and TCR-seq data have been deposited have been deposited in GEO with the accession codes GSE114595. Other data that support the findings of this study are available from the corresponding authors upon reasonable request.

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

## Acknowledgements

We thank C. Kim, L. Nosworthy, D. Hinz and R. Simmons for assisting with iNKT cell subsets sorting; J. Greenbaum, F. Zheng, A. Logandha R P and S. Kannan for help with processing and analyzing RNA-seq and TCR sequencing data; A. Denn for assisting with preparation of histology slides; S. Sakaguchi (Osaka University) and A. Weiss (UCSF) for SKG and ZAP70AS mice, respectively; A. Khurana (La Jolla Institute), G. Seo (La Jolla Institute), A. P. Uldrich (University of Melbourne), G. Nygaard (UCSD), C. Aguiar (University of São Paulo) for technical assistance. Supported by the US National Institutes of Health (AI71922 to M.K., R01AI070544 and R01AR066053 to N.B., T32 AR064194 fellowship to M.Z., S10OD016262 and S10RR027366 to La Jolla Institute, UL1TR001442 to UCSD), National Science Center grant (2016/23/B/NZ5/011469, Poland) and Broegelmann Foundation to P.M.

## Author contributions

M.Z., M.S., N.B., and M.K. designed the experiments. M.Z. and M.S. performed and analyzed the experiments. K.V. and D.E. provided human data. A.S. processed RNA-seq and TCRseq data. S.L. prepared libraries for RNA-seq. P.M. performed micro-CT

scanning. I.E. helped with experiments and provided helpful discussion. J.D. helped with TCRseq experiment. M.Z., M.S., N.B., and M.K. wrote the manuscript

## Additional information

**Competing interests:** The authors declare no competing interests.

