## [Peer Review File · Nature Communications]

REVIEWERS' COMMENTS:

Reviewer #1 (NKT, CD1d)(Remarks to the Author):

This manuscript has tested the hypothesis that TCR signaling strength during thymic selection influences the development of distinct subsets of invariant natural killer T (iNKT) cells. For this purpose, mutant alleles of the Zap70 signaling molecule are analyzed in mice, as well as a Zap70 inhibitor. Findings show that Zap70-deficiency or inhibition causes expansion of IFN-gamma-producing NKT1 cells, suggesting a reduced signaling threshold for these cells as compared with NKT2 and NKT17 cells. These NKT1 cells inhibited arthritis that spontaneously develops in one of the Zap70 mutant lines, thus showing biological relevance of the findings. Such NKT1 cells were also identified in the synovium of patients, suggesting a human correlate to the mouse studies. From these findings, it is concluded that TCR signaling strength during thymic differentiation influences iNKT cell cytokine production and the capacity of these cells to control disease.

General comments: This paper provides the most direct evidence to date that signaling strength during the intrathymic development of iNKT cells influences iNKT cell subset differentiation. A comprehensive set of studies is performed that includes proper controls and analyses.

Reviewer #2 (Thymic selection, TCR signaling)(Remarks to the Author):

This study analyses the effect of reduced TCR signaling on iNKT cell subsets. If Zap70 signaling is reduced using a hypomorphic allele or specific drug inhibition, then the subsets of NKT cells (iNKT1, 2, and 17) change. Specifically, NKT1 cells become more prevalent. A very comprehensive analysis of the changes is presented. The weaker signaling leading to increased NKT1 is interpreted as NKT1 cell development requiring weaker TCR signals than does development of NKT17 or NKT2 subsets – TCR signaling cascade genes are increased in expression in the NKT1 subset.

An impressive set of different experimental systems are brought to bear, including several genetic models for in vivo work, as well as FTOC for in vitro testing of drug inhibition. The ZAP70AS model that allows drug inhibition of Zap70 signaling is itself a hypomorph, showing a similar phenotype to the SKG mouse (more NKT1). When Zap70 signaling is inhibited by a drug, then the proportion of NKT2 cells declines, so the NKT2 cells are more sensitive to the weakened signal than the NKT1's, which naturally survive under the weaker TCR signaling regimen.

NKT1 cells in the SKG Zap70 hypomorphic mice have generally similar gene expression to WT NKT1 cells, but SKG NKT1 cells have lower expression of inhibitory receptors such as CD5, CD160, TIGIT, suggesting that the reduced TCR signaling in the SKG mice leads to a reduced amount of these molecules.

They show an interesting finding using a mannan-induced arthritis model in SKG mice. As the disease progresses, the number of NKT cells in the affected joints increases. Yet the disease is worse in the absence of iNKT cells, and ameliorated in their presence.

NKT1 cells producing IFN γ can prevent the severe form of arthritis, but their presence in the NKT population declines as the disease progresses, with a concomitant increase in NKT17, IL17 producing cells.